

# Air Quality Impact of the Northern California Camp Fire of November 2018

**Brigitte Rooney[1], Yuan Wang[1,2]\*, Jonathan H. Jiang[2], Bin Zhao[3], Zhao-Cheng Zeng[4], and John H. Seinfeld[5]\***

[1]Division of Geological and Planetary Sciences, California Institute of Technology, Pasadena, CA, USA

[2]Jet Propulsion Laboratory, California Institute of Technology, Pasadena, CA

[3]Pacific Northwest National Laboratory, Richland, WA, USA

[4]Joint Institute for Regional Earth System Science and Engineering, University of California, Los Angeles, CA, USA

[5]Division of Chemistry and Chemical Engineering, California Institute of Technology, Pasadena, CA, USA

\*Corresponding authors:

Yuan Wang: yuan.wang@caltech.edu

John Seinfeld: seinfeld@caltech.edu





**Abstract**

The Northern California Camp Fire that took place in November 2018 was one of the most damaging environmental events in California history. Here, we analyze ground-based station observations of airborne particulate matter that has a diameter < 2.5 micrometers ($PM_{2.5}$) across northern California and conduct numerical simulations of the Camp Fire using the Weather Research and Forecasting model online coupled with Chemistry (WRF-Chem).

Simulations are evaluated against ground-based observations of $PM_{2.5}$, black carbon, and meteorology, as well as satellite measurements, such as Tropospheric Monitoring Instrument (TROPOMI) aerosol layer height and aerosol index. The Camp Fire led to an increase in Bay Area $PM_{2.5}$ to over 70 µg m-3 for nearly two weeks, with localized peaks exceeding 300 µg m-3. Using the Visible Infrared Imaging Radiometer Suite (VIIRS) high resolution fire detection products, the simulations reproduce the magnitude and evolution of surface $PM_{2.5}$ concentrations, especially

downwind of the wildfire. The overall spatial patterns of simulated aerosol plumes and their heights are comparable with the latest satellite products from TROPOMI. WRF-Chem sensitivity simulations are carried out to analyze uncertainties that arise from fire emissions, meteorological conditions, feedback of aerosol radiative effects on meteorology, and various physical parameterizations, including the planetary boundary layer model and the plume rise model. Downwind $PM_{2.5}$ concentrations are sensitive to both flaming and smoldering emissions over the fire, so

the uncertainty in the satellite derived fire emission products can directly affect the air pollution simulations downwind. Our analysis also shows the importance of land surface and boundary layer parameterization in the fire simulation, which can result in large variations in magnitude and trend of surface $PM_{2.5}$. Inclusion of aerosol radiative feedback moderately improves $PM_{2.5}$ simulations, especially over the most polluted days. Results of this study can assist in the development of data assimilation systems as well as air quality forecasting of health exposures and

economic impact studies.

**1 Introduction**

Wildfires have become increasingly prevalent in California. It has been reported that between 2007 and 2016, as many as 3672 fires occurred in California, consuming up to 434,667 acres (Pimlott et al., 2016). Increasingly, the population has expanded into high fire-risk areas and near wildland-urban interfaces (Brown et al., 2020). The intense

smoke consisting of airborne particulate matter of diameter < 2.5 micrometers ($PM_{2.5}$) associated with these fires leads to an increased risk of morbidity and mortality (Cascio, 2018). $PM_{2.5}$ from wildfires consists of a spectrum of light scattering and absorptive particles largely comprising organic and black carbon. It is increasingly important to understand the cause and nature of wildfires as the number of extreme events and the length of the wildfire season continue to grow (Kahn, 2020; Shi et al., 2019). Fire-related studies have estimated exposures to $PM_{2.5}$ based on

ground-level monitoring-station measurements (Shi et al., 2019; Herron-Thorpe et al., 2014; Archer-Nicholls et al., 2015). Spatial coverage of such monitoring stations often tends to be scarce, especially in rural areas. Satellite remote sensing offers a powerful method to monitor air quality during fire events. One study used radiance measurements from the TROPOspheric Monitoring Instrument (TROPOMI) to derive atmospheric carbon monoxide and assess the resulting air quality burden in major cities due to emissions from the California wildfires from November 2018

(Schneising, et al., 2020). Ideally, analysis of fire events is based on a combination of satellite-based measurements





and ground-level observations to obtain spatial and temporal distributions of emissions. The Camp Fire of November 2018 was, to date, the deadliest and most destructive wildfire in California (Kahn, 2020; Brown et al., 2020). Originating along the Sierra Nevada mountain range, smoke from the fire spread across the Sacramento Valley to the San Francisco Bay Area. Peak levels of $PM_{2.5}$ in the San Francisco area exceeded 200 µg m$_{-3}$ and remained above 50

µg m$_{-3}$ for nearly two weeks.

Numerous studies have addressed wildfire events using a variety of model frameworks and data sources (Shi et al., 2019; Herron-Thorpe et al., 2014; Archer-Nicholls et al., 2015; Sessions et al., 2011).  Shi et al. (2019) used the WRF-Chem model with Moderate Resolution Imaging Spectroradiometer (MODIS) and VIIRS fire data to study the wildfire of December 2017 in Southern California. Herron-Thorpe et al. (2014) evaluated simulations of

the wildfires in the Pacific Northwest of 2007 and 2008 using the Community Multi-scale Air Quality (CMAQ) model with fire emissions generated by the BlueSky framework and fire locations determined by the Satellite Mapping Automated Reanalysis Tool for Fire Incident Reconciliation (SMART-FIRE). That study suggested that underprediction of $PM_{2.5}$ was the result of underestimated burned area as well as underpredicted secondary organic aerosol (SOA) production and incomplete speciation of SOA precursors within the CMAQ model. Archer-Nicholls et

al. (2015) simulated biomass burning aerosol during the 2012 dry season in Brazil using WRF-Chem and fire emissions prepared from MODIS. That study proposed that biases in the model were likely a result of uncertainty in the plume injection height and emissions inventory, as well as simulated aerosol sinks (e.g., wet deposition), and lack of inclusion of SOA production in the Model for Simulating Aerosol Interactions and Chemistry (MOSAIC). Sessions et al. (2011) investigated methods for injecting wildfire emissions using WRF-Chem. That study tested two fire data

preprocessors: PREP-CHEM-SRC (included with WRF-Chem) and the Naval Research Laboratory's Fire Locating and Monitoring of Burning Emissions (FLAMBE), and three injection methods: the 1-D plume rise model within WRF-Chem, releasing emissions only within the planetary boundary layer, and releasing emissions between 3 and 5 km. That study compared results from simulating wildfires during the NASA Arctic Research of the Composition of the Troposphere from Aircraft and Satellites (ARCTAS) field campaign in 2008 with satellite data. Sessions et al.

(2011) found that differences in injection heights result in different transport pathways.

The present study is a comprehensive investigation of air quality impacts of the Camp Fire using a combined analysis of ground-based and space-borne observations and WRF-Chem simulations. Descriptions of the observation and model are presented in Section 2; model evaluation is presented in Section 3; results of analysis are given in Section 4, followed by discussion and conclusion in Section 5.

**2 Model Description and Observational Data**

The present study employs WRF-Chem (version 3.8.1) driven by the latest version of meteorological reanalysis data for initialization and boundary conditions. Fire emissions are determined by pairing active fire location data from VIIRS Satellite with the Brazilian Biomass Burning Emission Model (3BEM), which calculates species mass emissions from the burned biomass carbon density, combustion factors, emission factors, and the burning area. WRF-

Chem simulations are evaluated against EPA surface observations and TROPOMI satellite products.



## 2.1 WRF-Chem Configuration

The WRF-Chem simulation time period is 7 November 2018 (a day before the fire began) to 22 November 2018 (when the fire was 90% contained). We carried out simulations over two domains (Fig. 1): Domain 1 includes all of California at 8 km × 8 km horizontal resolution, while Domain 2 covers Northern California at 2 km × 2 km horizontal resolution. 49 vertical layers are used from the surface to 100 hPa with 50 m vertical resolution in the planetary boundary layer. The meteorological boundary and initial conditions for the outer domain are generated from the fifth generation of European Centre for Medium-range Weather Forecasts (ECMWF) Re-Analysis dataset (ERA5) at 30 km × 30 km resolution (Copernicus Climate Change Service, 2017). Chemical boundary and initial conditions for the outer domain are generated from the Model for Ozone and Related Chemical Tracers version 4 (MOZART-4) (University Corporation for Atmospheric Research, 2013).

We use physical options of the Noah Land-Surface Model (Tewari et al., 2004), the Mellor-Yamada-Janjic (MYJ) boundary layer scheme (Janjic, 1994), and the RRTM (longwave) and Dudhia (shortwave) radiative transfer schemes (Dudhia, 1989). Cumulus parameterization is not included. The second-generation Regional Acid Deposition Model (RADM2) chemical mechanism coupled with the Modal Aerosol Dynamics model for Europe (MADE) and Secondary Organic Aerosol Model (SORGAM) (Zhao et al., 2011) are employed. Aerosol optical properties are calculated based on the volume approximation, for which the volume average of each aerosol species is used to calculate refractive indices (Jin, et al., 2015). Aerosol radiative feedbacks on meteorology and chemistry are included in the simulations.

We use the National Emission Inventory for anthropogenic emissions (US EPA, 2018). Biogenic emissions are calculated online using the Guenther scheme (Guenther et. al., 2006). Dust emissions are calculated online using the Goddard Chemical Aerosol Radiation Transport (GOCART) dust emission scheme with University of Cologne (UOC) modifications (Shao et al., 2011). Sea salt emissions are excluded. Technical details of wildfire emissions and the plume rise calculation are discussed in the next section.

## 2.2 Fire Emissions Inventory and Plume Rise Model

Wildfire emissions are generated using the PREP-CHEM-SRC v1.5 preprocessor (Freitas et al., 2011) employing the Brazilian Biomass Burning Emission Model (3BEM, Longo et al., 2010) with satellite data on detected fires. For each pixel with fire detected, the mass of emitted species is calculated by:

$$M^{[\eta]} = \alpha_{veg} \cdot \beta_{veg} \cdot EF_{veg}^{[\eta]} \cdot a_{fire} \tag{1}$$

for a certain species η, where $\alpha_{veg}$ is the carbon density (the mass of burnable above-ground biomass per unit area of vegetation), $\beta_{veg}$ is the combustion factor, $EF_{veg}$ is the emission factor by species and vegetation type, and $a_{fire}$ is the burning area of each fire pixel. Vegetation type is generated from the MODIS data following IGBP land cover classification. Vegetation type-specific emission factors ($EF_{veg}$) and combustion factors ($\beta_{veg}$) are derived from Ward et al. (1992) and Andreae and Merlet (2001). Vegetation type-specific carbon density ($\alpha_{veg}$) is based on Olson et al. (2000) and Houghton et al. (2001). Active fire detection is retrieved from the VIIRS fire product with 375 m spatial



resolution. A limitation of the VIIRS fire count product is its relatively low temporal resolution. As a polar-orbiting satellite, VIIRS provides fire detection during the daytime only once (about 13:30 local time) at each location.

The emission preprocessor generates a file formatted for WRF-Chem containing the smoldering-phase surface emission fluxes of each species, the fire size for each vegetation type, and flaming factor. Flaming factor is the ratio of biomass consumed in the flaming phase to biomass consumed in the smoldering phase. The 17 IGBP

land cover classes are aggregated into four main types: tropical forest, extratropical forest, savanna, and grassland. The size of the wildfire and phase of combustion play important roles in the structure of the plume and the vertical distribution of emissions. Wildfire combustion is generally considered to occur in two phases: smoldering and flaming. Emissions from the smoldering phase are allotted to the first layer of the computational grid, while those from the flaming phase are released at injection heights above the surface, as determined by the plume rise model

described below. Fire size determines the total surface heat flux, as well as the entrainment radius of the plume. Fire parameters are ascribed a daily temporal resolution and are distributed to the WRF-Chem domains. The fire parameters are then input to the plume rise model (Freitas et al., 2007, 2010). The plume rise model is a 1-dimensional model implemented in each WRF-Chem grid cell with an independent vertical grid resolution of 100 m. It calculates the maximum height to which a plume reaches and distributes emissions therein (Fig. 2). The plume top

height, determined by the surface heat flux from the fire and the thermodynamic stability of the atmospheric environment, is defined as the height at which the in-plume parcel vertical velocity $< 1$ m s$^{-1}$. The plume rise model uses upper and lower bounds of heat fluxes determined by each land type to calculate the minimum and maximum plume top height. Flaming emissions are distributed equally to each vertical level within the injection layer with the following calculation: *Flaming Emission per Level = Smoldering Emission × Flaming Factor × DZ$^{-1}$*, where *DZ =*

*Maximum Plume Top Height – Minimum Plume Top Height*. The model also accounts for entrainment, water balance, and internal gravity wave damping.

Figure 3 shows the fire size and particulate matter emissions produced from MODIS and VIIRS data. The Camp Fire burned primarily extratropical forest vegetation (which comprised 68% of the total burned area), followed by savanna (23% of total area). The flaming emission rate for species *n* from vegetation type *v*, is calculated by

$$Flaming\ Phase\ Rate_{n,v} = \sum\nolimits_{Fire\ Cells} Area_v \cdot Smoldering\ Phase\ Flux_n \cdot Flaming\ Factor_v \qquad (2)$$

At maximum, the carbon monoxide (CO) emission flux was $4.1 \times 10^7$ mol km$^{-2}$ hr$^{-1}$, and PM$_{2.5}$ flux was $3.7 \times 10^4$ µg m$^{-2}$ s$^{-1}$. On average, 46% of the fuel burned is estimated to have been consumed during the flaming phase.

Fire Inventory from NCAR (FINN) Version 1.5 (Wiedinmyer, 2011) is another fire emissions product that we will test in a sensitivity analysis. It is assembled for atmospheric chemistry models with a daily temporal resolution

and a 1 km horizontal resolution. FINN is generated using satellite observations of active fires and land cover paired with emission factors and fuel loading estimates. The emissions are allocated to a diurnal cycle following WRAP (2005). FINN outputs the total wildfire emission flux, fire size, and land type fraction. As FINN does not include a smoldering-to-flaming phase ratio, the plume rise model calculates a ratio based on CO emissions.

**2.3 Surface and Satellite Observations**



The observational data include both ground-based measurements and satellite observations. Meteorological and surface concentration data were obtained from the NOAA's National Climatic Data Center (NCDC) and EPA Air Quality System (AQS), respectively. We focus on three areas: the region closest to the fire, the Sacramento Metro Area (population of 2.5 million), and the San Francisco Bay Area (population of 7 million). Hourly observations of wind speed at 10 m, wind direction at 10 m, temperature at 2 m, $PM_{2.5}$, black carbon (BC), and CO are available for the sites shown in Fig. 1. We use level-2 products from the TROPOMI onboard the Copernicus Sentinel-5 Precursor satellite (S5P) to evaluate the spatial and vertical distribution of predictions. We compare TROPOMI aerosol layer height retrievals (3.5 km × 7 km) with the predicted WRF-Chem height of maximum $PM_{2.5}$, and ultraviolet aerosol index (UVAI, 3.5 km × 7 km) with the predicted WRF-Chem BC columns. The model results are sampled around 13:30 local time when S5P passes over California.

**2.4 Control and Sensitivity Simulations**

To investigate the effects of key model parameters on the ability to predict the atmospheric impact of the wildfire, we conduct a range of sensitivity simulations. As meteorology and atmospheric structure play important roles in plume dynamics and the transport of particulate matter, we separately perturb the aerosol radiative feedback to meteorology, the planetary boundary layer parameterization, and the plume entrainment coefficient. To understand further the extent to which fire characteristics provided by satellite data can affect the simulations, we analyze the influence of fire data sources, the emission rate, and partitioning between smoldering phase and flaming phase emissions. A summary of these simulations is provided in Table 1.

Our evaluation focuses on the control simulation (S_CTRL). S_CTRL applies a factor of 3 to the smoldering emissions on 13 November and a factor of 2 to the smoldering emissions on 14-16 November due to the intermittent cloudy conditions over the northern California on those days. S_CTRL uses the native flaming factor and fire size products, the default entrainment constant of 0.05, and the Mellor-Yamada-Janjic planetary boundary layer scheme. In the following scenarios, one parameter is individually perturbed from this configuration. S_EMRAW uses the native emissions input with unaltered smoldering phase emissions, S_NOAERO turns off the aerosol radiative feedback to meteorological fields, S_FCTX2 doubles the flame factor for the entire simulation period (thus increasing flaming phase emissions without changing the smoldering phase), S_ENTR reduces the entrainment coefficient within the plume rise model from 0.05 to 0.02, and S_LSM employs an alternative land surface model and planetary boundary layer scheme. We perform another sensitivity simulation using FINN in place of VIIRS (S_FINN).

**3 Evaluation of Fire Simulations**

**3.1 Meteorology**

The three spatial areas of our interest differ significantly in topography and meteorology. Figure 4 shows the averaged wind observations and S_CTRL predictions. S_CTRL captures general wind patterns and achieves strong correlation with observed temperatures in each of the areas (Fig. 5). In the first few days of the Camp Fire, the foothills



and the Sacramento area experienced strong northerly winds, while the Bay Area experienced northeasterly winds, both predicted by the simulation. Other distinct features like those on 11 November near the fire and in the Bay Area are also reproduced by S_CTRL with some bias in timing. In the Bay Area, winds were typically southerly at speeds less than 2 m s-1 and consistent through most of the simulation duration. In the relatively dry Sacramento Valley inland, winds were also predominantly southerly, but were calmer (< 1 m s-1) and varied more than those on the coast. After 11 November, the wind speeds were much slower. Coastal air regulates Bay Area temperatures, whereas the drier Sacramento area experiences a greater temperature range. S_CTRL also produced these relative characteristics, but, in general, generated faster winds and higher temperatures than those observed. A summary of model performance statistics is provided in Table 2. The complex terrain of the Bay Area and the Sierra Nevada Foothills near the fire location likely contribute to uncertainty in predicting meteorological parameters.

### 3.2 Surface-Level Particulate Matter

Figure 6 shows the predicted evolution of surface PM2.5 from AQS observations and S_CTRL over the period of the wildfire. Within hours of the onset of the Camp Fire, observed PM2.5 concentrations in Sacramento and the San Francisco Bay Area (130 and 240 km downwind) increased from below the National Ambient Air Quality Standard (NAAQS) 24-h average of 35 µg m-3 to 50 µg m-3. Both areas remained above the standard for more than a week, reaching values of three times the standard for multiple days. The region near the fire, Sacramento, and the San Francisco Bay Area were each out of attainment of the NAAQS 24-h average of PM2.5 for 11, 11, and 12 days, respectively, during 7-20 November, while S_CTRL predicted 12, 11, and 11 days, respectively. Much of northern California did not return to attainment until 22 November when the wildfire reached 90% containment. Table 3 summarizes the ability of S_CTRL to reproduce observed values of surface PM2.5 in the three focus areas and at stations 27 and 28 in the Bay Area. The model prediction exhibits a mean bias of 64.8 µg m-3 in the region of the Camp Fire, -11.4 µg m-3 in Sacramento, and -16.8 µg m-3 in the Bay Area. Mean bias was smaller at some individual monitoring stations, such as Station 27 and 28 that has mean bias of -9.9 µg m-3 and -6.2 µg m-3, respectively. In the broader area near the fire, S_CTRL significantly overestimates surface PM2.5, reaching nearly 1 mg m-3 while observed concentrations peaked closer to 300 ug m-3. However, S_CTRL shows a similar temporal trend to that observed, capturing many peak times. The Sacramento area experienced maxima near 300 µg m-3, while the Bay Area reached around 200 µg m-3. S_CTRL shows good agreement of the magnitude and temporal evolution of surface PM2.5 in the Bay Area and Sacramento for most days, with the exception of 10 November and 14-16 November (to be discussed subsequently). Time series of observed and predicted surface CO and BC in the Bay Area are shown in Fig. 7. Again, S_CTRL shows good agreement with the magnitude and trend of both species. While PM2.5 is largely underpredicted in the period of 14-16 November, BC is over predicted by 5-10 µg m-3 at peaks. S_CTRL also produces positive bias in surface CO over 16-18 November.

Error in surface PM2.5 can, in part, be attributed to error in the predicted wind fields. In the latter hours of 8 November near the Camp Fire, S_CTRL predicts southerly winds, while observations are steadily northerly, leading to some return of initially transported plume. Again, on 11 November, predicted winds show a dramatic reversal, and surface PM2.5 spikes. In Sacramento on 10 November, observed and predicted northerly winds at midday initially lead





to increased PM$_{2.5}$ concentrations, but winds swing southerly in the later hours. On 13 November, observed winds
blow south and transport emissions to Sacramento, while S_CTRL predicts winds in the opposing direction, leading
to an underprediction in PM$_{2.5}$. However, error in predicted wind fields does not explain the substantial
underprediction of surface PM$_{2.5}$ in the Bay Area over 14-16 November, as the station-averaged winds of the area do
not show significant deviation from observations. We tested the Four-Dimensional Data Assimilation (FDDA) of
large-scale horizontal wind from the ERA5, but it could not reduce the aforementioned biases in wind, possibly due
to the fact that the observed wind patterns are driven by some mesoscale or even local-scale dynamics.

To study the structural evolution of the wildfire plume, we compare simulated total black carbon column with
TROPOMI UVAI satellite retrievals (Fig. 8). TROPOMI UVAI is based on the difference between wavelength-
dependent Rayleigh scattering observed in an atmosphere with aerosols and that of a modeled molecular atmosphere
(Stein Zweers et al., 2018). This difference is measured in the UV spectral range where ozone absorption is small. A
positive residual (red coloring) indicates the presence of UV-absorbing aerosols, like black carbon (BC), while a
negative residual (blue coloring) indicates presence of non-absorbing aerosols. As WRF-Chem does not generate an
aerosol index parameter, we compare UVAI to total BC column, a significantly absorbing aerosol. Over the period of
the simulation, broad characteristics and shape, as well as some more distinct features, of the Camp Fire plume are
reproduced by S_CTRL. Using similar input data sources and WRF-Chem configuration, but a simpler plume rise
model, Shi et al. (2019) also capture the general shape of the plume, but underestimate aerosol magnitude.
Discrepancies in S_CTRL plume transport correlate to bias in surface PM$_{2.5}$. On the first day of the fire, observations
show that strong winds in northern California drag the plume west, where steady coastal winds transported the plume
south and inland again (Fig. 8). The dynamics creates a dense plume with two narrow stretches. S_CTRL predictions
of total BC column fail to capture the hook-shape present in the UVAI retrievals but reflect the two separate stretches
of narrow plume. The simulation constrains one stretch to the valley, leading to overprediction of surface PM$_{2.5}$ in
Sacramento on 8 November (Fig. 6b). On 11 November, the simulation does not reproduce the second band of the
plume which wraps along the coast and towards San Francisco; rather, the plume remains more concentrated to the
Sacramento Valley again. This leads to underprediction of surface PM$_{2.5}$ in the Bay Area and overprediction in
Sacramento (Fig. 6b and c). The narrow PM$_{2.5}$ peaks of S_CTRL on 14-16 November in Sacramento can likely be
attributed to the more pronounced plume on 14 November and 16 November. A stark horizontal gradient of fire
emissions could restrict accumulation of PM$_{2.5}$ averaged over the Sacramento region.

To investigate the predicted decrease of surface PM$_{2.5}$ in the Bay Area in the afternoon of 14 November, we
individually analyze station 27 (Fig. 9) and station 28 (Fig. 10). Figures 9 and 10 show the vertical profile of S_CTRL
PM$_{2.5}$ concentrations, the observed and predicted surface PM$_{2.5}$, and the observed and predicted wind fields.
Additionally, Fig. 11 shows the spatial distribution of PM$_{2.5}$ and surface winds of observations (a) and predictions (b)
at four times on 14 November. In the late morning at station 27, observed winds become northeasterly and PM$_{2.5}$
spikes as more particle-laden air flows westward (Fig. 9). At the same time, S_CTRL winds also become northeasterly
and PM$_{2.5}$ increases accordingly. However, predicted winds reverse, and PM$_{2.5}$ levels remain relatively low from
midday 14 November to midday 15 November. Station 28 exhibits similar behavior of an increase in PM$_{2.5}$ with wind



change, then a sharp drop as predicted winds deviate strongly northward. This behavior emerges as part of a larger flow pattern in Fig. 12. Throughout the morning of 14 November, the simulated wildfire plume approaches the Bay Area and is then driven back inland by a strong sea breeze in the afternoon, not present in the observational data. This behavior is also demonstrated in the vertical profile of PM2.5 (Fig. 9a and 10a). A column of clean air flushing the Bay Area leads to a predicted bias of -50 µg m-3 on 15 November.

**3.3 Aerosol Vertical Profile**

    The TROPOMI ALH retrieval represents vertically localized aerosol layers within the free troposphere in cloud-free conditions and is designed to capture aerosol layers produced by biomass burning aerosol (such as wildfires), volcanic ash, and desert dust (Apituley et al., 2019). ALH is retrieved based on the significant effect of aerosol vertical structure on the high spectral resolution observations in the $O_2$-A band in the near-infrared (759 to 770 nm). The ALH
algorithm includes a spectral fit estimation of reflectance across the $O_2$ A band using the Optimal Estimation retrieval method with primary fit parameters of aerosol layer mid pressure and aerosol optical thickness (de Graaf et al., 2019). The assumed aerosol profile is a single uniform scattering layer with a fixed pressure thickness, constant aerosol volume extinction coefficient, and constant aerosol single scatter albedo. The mid pressure of the layer, defined as the average of the top and bottom pressures, is converted to altitude with a temperature profile. This parameterization is
best suited for aerosol profiles dominated by a sole elevated and optically thick aerosol layer, which is characteristic of wildfire plumes.

    We compare the satellite-derived aerosol layer height to WRF-Chem predictions of PM2.5 using two methods. We define the smoke aerosol layer with a PM2.5 threshold concentration of 3 µg m-3. For the first method, the layer height is calculated as the average of heights at which PM2.5 is greater than the threshold. For the second method, these
heights are weighted by BC mass. Figure 12 shows the satellite-derived layer height (a) and the S_CTRL model bias of average heights (b) and mass weighted average heights (c). TROPOMI layer heights are generally 1 to 2 km and reach greater than 6 km in some instances. Using purely averaged heights, S_CTRL typically overpredicts ALH by 100 to 400 m and remains within a smaller range than TROPOMI. S_CTRL layer heights weighted by BC mass are lower, thus improving agreement with the satellite. Note that the reported retrieval bias in TROPOMI ALH is about
780 m for wildfire emission plumes (Nanda et al., 2020). Archer-Nicholls et al. (2015) and Sessions et al. (2011) also reported overpredicted aerosol layer heights using WRF-Chem when compared to airborne data and Multi-angle Imaging Spectro Radiometer MISR stereo heights, respectively. Using CMAQ, however, Herron-Thorpe et al. (2014) reported underpredicted heights when compared to Cloud-Aerosol Lidar with Orthogonal Polarization CALIOP products. Archer-Nicholls et al. (2015) found that error in plume injection height can contribute to error in surface
PM, and that PM biases were dependent on vegetation type as carbon-density and heat release vary by vegetation. Location of the aerosol layer within the column likely also contributes to error in surface predictions of PM2.5 in this study, however, the current analysis is inconclusive. The assumption of a single, elevated aerosol layer used in the TROPOMI ALH derivation may not be characteristic of the vertical structure predicted by WRF-Chem. As seen in Fig. 9 and 10 and in the vertical profile near the wildfire, layers of aerosol are commonly present at the surface and
exist as multiple nonlocalized layers. Sessions et al. (2011) also found that using the FLAMBE fire data preprocessor

with emission injection heights not constrained to the boundary layer resulted in better agreement with satellite products than PREP-CHEM-SRC. Consideration of the WRF vertical grid is also necessary when comparing surface level values. Further development of the analytic method used to evaluate WRF-Chem aerosol layer heights may provide insight into the behavior of the plume rise model and its vertical structure.

## 4 Sensitivity Simulation Analysis

We conduct sensitivity simulations to investigate the effects of various parameters on the ability of the WRF-Chem model to accurately predict downwind PM concentrations from wildfires. As meteorological conditions and related boundary structure play important roles in plume dynamics and the transport of PM, we separately test the aerosol feedback to meteorology and the land surface model. To understand the extent to which fire characteristics provided by satellite data can affect the simulation, we analyze the fire product sources (VIIRS versus FINN), the total fire emissions, and the division between smoldering versus flaming phase emissions. To examine the influence of the plume rise model, we perturb a key parameter, the entrainment coefficient.

### 4.1 Aerosol Radiative Feedback to Meteorology

By absorbing and scattering solar radiation, aerosols can impact the radiative fluxes, cloud formation, and precipitation in the atmosphere (Wang et al., 2016; 2020), and, in turn, the meteorological conditions for aerosol formation, transport, and removal (Li et al., 2019). WRF-Chem has the option to couple aerosol-radiative direct effects with meteorology simulation. S_NOAERO uses the same input data and configuration as S_CTRL, but disables the aerosol radiative feedback. Figure 13 shows the evolution of surface wind speed and temperature throughout the wildfire near the source (a), in Sacramento (b), and in the Bay Area (c). The aerosol radiative impact on simulated meteorology is more pronounced for surface temperature than wind. When aerosol radiative feedbacks are noticeable, colder temperatures and calmer winds are found near the surface. Generally, feedbacks are more evident in the region closer to the fire sources with larger PM concentrations. Also, in the Bay Area, the largest changes in meteorology coincide with the largest differences in surface $PM_{2.5}$ between the two scenarios (Fig. 14), which occurs when higher concentrations are predicted (10-11 November, 14-16 November). Consequently, the aerosol radiative feedback in WRF-Chem acts to stabilize the atmosphere, presumably due to the solar absorption by smoke aerosols and reduction of radiation reaching the surface (Wang et al., 2013). When taking the entire time period into account, the overall aerosol effect on meteorology is relatively small in the downwind region, like the Bay Area, even when aerosol concentrations are high.

### 4.2 Fire Emission Inventory

WRF-Chem input fire files produced with VIIRS and PREP-CHEM-SRC include fire size, smoldering emission flux, and flaming factor. Here, we test the sensitivity of predictions to FINN (S_FINN) versus VIIRS/MODIS, as well as the smoldering emission flux (S_EMRAW) and flaming factor (S_FCTX2). S_FINN produces very little aerosol, though it captures the timing of some peaks. The aerosol underestimation may be a result of bias in the emission

inventory or an issue of its implementation in the plume rise model code, as FINN specifies total wildfire emissions
rather than a smoldering and flaming distribution.

When VIIRS emission inventory is used, the total wildfire emission flux can be altered through two parameters: the smoldering emission flux at the surface and the flaming factor. Directly increasing the smoldering emission flux adds emissions to the surface layer and increases flaming phase emissions proportionally. Figure 14 shows the impact of doubling smoldering emissions on 13 November and tripling them during 14-16 November. These changes to the
inventory more than double concentrations of surface $PM_{2.5}$ in the area of the wildfire and increase concentrations in the Bay Area by 20 to 60 µg $m_{-3}$ during 14-16 November. Consequently, increasing input of total wildfire emissions improves the agreement of predictions with observations in Sacramento and the Bay Area, suggesting that some uncertainty may stem from satellite fire products. This finding is supported by Archer-Nicholls et al. (2015), as they applied a factor of 5 to scale up the wildfire emissions in their simulations. By modifying the flaming factor, we
perturb only the emissions injected aloft by the plume, as emissions higher in the atmosphere may allow for greater transport downwind. By doubling the flaming factor over the full simulation duration, S_FCTX2 recovers 10-35 µg $m_{-3}$ in the Bay Area 14-16 November (Fig. 14c), when S_CTRL substantially underpredicts $PM_{2.5}$.

### 4.3 Plume Rise Parameterization – Entrainment Coefficient

The plume rise model parameterizes entrainment as proportional to the plume vertical velocity and inversely
proportional to the plume radius (Freitas et al., 2010). Greater entrainment causes rapid cooling, such that near surface plume temperatures are only slightly warmer than the environment, lowering buoyancy and reducing the plume height. Larger wildfires generate less entrainment and reach higher injection heights. The parameterization also includes the effect of horizontal winds on entrainment. Strong wind shear can enhance entrainment and increase boundary layer mixing (Freitas et al., 2010). Archer-Nicholls et al. (2015) decreased the original entrainment coefficient (Freitas et
al., 2007) from 0.1 to 0.05 to improve their simulations of a wildfire. As the Camp Fire developed rapidly and intensely, we performed the sensitivity simulation S_ENTR with a lower entrainment coefficient of 0.02 to allow for higher injection heights. However, entrainment perturbation resulted in less than 1% change in surface $PM_{2.5}$ from S_CTRL. A possible reason is that the background winds were quite strong already, for which the entrainment coefficient played a limited role.

We compare simulations using two different land surface models (LSM) which include the PBL schemes: the Noah LSM with Mellor-Yamada-Janjic (MYJ) PBL and the Pleim-Xiu LSM (referred to here as P-X) with the Asymmetric Convection Model 2 (ACM2) PBL (Janjic, 1994; Pleim and Xiu, 1995; Chen & Dudhia, 2001; Pleim, 2007). Land surface models simulate the heat and radiative fluxes between the ground and the atmosphere (Campbell et al., 2018). Noah LSM has four soil moisture and temperature layers, while the Pleim-Xiu LSM has two (Hu et al,
2014; Campbell et al., 2018). Both include a vegetation canopy model and vegetative evapotranspiration. The PBL scheme provides the boundary layer fluxes (heat, moisture, and momentum) and the vertical diffusion within the column. It uses boundary layer eddy fluxes to distribute surface fluxes and grows the PBL by entrainment. A key feature of PBL schemes is the inclusion of local mixing (between adjacent layers) and/or nonlocal mixing (from the





surface layer to higher layers). The MYJ scheme is a turbulent kinetic energy prediction, while the ACM2 scheme is
a member of the diagnostic non-local class. MYJ solves for the total kinetic energy in each column from buoyancy
and shear production, dissipation, and vertical mixing. ACM2 has two main components: a term for local transport by
small eddies and a term for nonlocal transport by large eddies. Coniglio et al. (2013) showed that the MYJ scheme
can undermix the PBL in locations upstream of convection in the presence of overly cool and moist conditions near
the ground in the daytime, whereas ACM2 can result in an excessively deep PBL in evening. Pleim (AMS, 2007) also
noted that ACM2 predicts the PBL profile of potential temperature and velocity with greater accuracy.

The use of P-X and ACM2 results in substantially different aerosol trends and plume evolution, the effects of
which are largely location-dependent (Fig. 14). Near the fire and in the Bay Area, S_LSM produces little similarity in
surface PM$_{2.5}$ magnitude and trend as compared to S_CTRL. S_LSM reduces PM$_{2.5}$ concentrations by more than 50%
in both areas for the majority of the simulation period. However, S_CTRL overpredicts PM$_{2.5}$ near the wildfire, while
S_LSM underpredicts but produces a more muted temporal pattern, similar to observations. In the Sacramento area,
S_LSM generally predicts higher PM$_{2.5}$ values with a distinct diurnal trend. Peaks are of similar magnitude to
S_CTRL, but displaced temporally. The topography of the Sacramento area is more uniform than the complex terrain
of the Bay area as well as the foothills and canyons near the wildfire, likely contributing to the distinctions in the
behavior of the two schemes. Moreover, the current sensitivity study stresses the importance of the parameterization
of the land surface and the boundary layer. As shown here, the Noah LSM and MYJ scheme performs well for the
broader region of northern California, whereas improvement near the wildfire itself may be attained with altered PBL
parameterization.

## 5 Conclusions and Discussion

The record-breaking Camp Fire ravaged northern California for nearly two weeks. At a distance of 240 km
downwind of the wildfire, Bay Area surface PM$_{2.5}$ levels reached nearly 200 µg m$_{-3}$ and remained over 70 µg m$_{-3}$ over
7-22 November 2018. It is uncertain to what extent the current chemical transport models can reproduce the key
features of this historical event. Here, we employ the WRF-Chem model to characterize the spatio-temporal PM
concentrations across northern California and to investigate the sensitivity of predictions to key parameters of the
model. The model utilizes satellite fire detection products with a resolution of 375 m and a biomass burning model to
generate the fire emission inventory at near real time. We conduct model simulations at 2 km resolution. A wide range
of observational data is employed to evaluate the model performance, including ground-based observations of PM$_{2.5}$,
black carbon, and meteorology from EPA and NOAA stations, as well as satellite measurements, such as Tropospheric
Monitoring Instrument (TROPOMI) aerosol layer height and aerosol index.

We focus on three geographic areas: the vicinity of the wildfire, Sacramento, and the San Francisco Bay Area.
The control experiment was able to simulate the general transport and extent of the plume as well as the magnitude
and temporal evolution of surface PM$_{2.5}$ in Sacramento and the Bay Area. Meanwhile, the control experiment
substantially overpredicted surface PM$_{2.5}$ near the fire, but captured the general evolution of the fire development. On
the Pacific coast, the Bay Area was subject to significant sea breezes not observed during the time period of simulation.



Owing to strong winds predicted from the ocean, a large negative bias existed in surface $PM_{2.5}$. Increasing total wildfire
emissions (smoldering + flaming) and increasing flaming phase emissions alone each recovered some $PM_{2.5}$ biases.
Aerosol radiative feedback on meteorology acted to stabilize the atmosphere and slightly increased the $PM_{2.5}$
concentration near the surface during most severe episodes. Hence, its inclusion modestly improves model
performance. Our study shows that sources of downwind PM error stem primarily from the localized structure of the
plume and uncertainty in fire emissions. Uncertainty of partitioning between smoldering and flaming phases may also
contribute to uncertainty in plume horizontal transport.

Future studies are needed to further improve the present modeling framework to simulate wildfires. Some
wildfires exhibit a distinct diurnal cycle, but the current fire preparation module has not utilized the nighttime fire
radiative power measurements by the polar-orbiting satellites. Also, the current land cover and vegetation type data
are still relatively coarse in spatial resolution and classification accuracy, which cannot fully resolve a small town in
a rural area. In fact, the Camp Fire reportedly burned the town of Paradise, California between 8 and 10 November
2018. This discrepancy definitely contributes to the uncertainty in the fire emission preparation. Additional
verification of input fire data sources, such as FINN, and their implementation in the WRF-Chem plume rise model is
needed for studies of the vertical structure. Deeper understanding of the role of plume dynamics and boundary layer
parameterization on aerosol concentrations downwind from wildfires will inform updates to forecast models like
WRF-SFIRE-CHEM, which couples WRF with a fire spread model and smoke dispersion simulation (Barbunzo 2019;
Kochanski et al., 2013). Given the complexity of the problem, here we only perturb individual factors in this study.
Future studies can test different combinations of the main factors identified by the present study, which can yield
additional insights about non-linear interactions among different processes related with fire emission and transport.

The recent TROPOMI aerosol layer height product shows promise as an analytical tool, but requires further
development of the method by which it can be directly compared to WRF-Chem. Given the assumptions required to
perform the TROPOMI ALH retrieval, more research is needed to compare that product with any height retrievals
from MODIS/MAIAC (Lyapustin et al. 2019), MISR, and CALIPSO. The intercomparison can help quantify
measurement uncertainty. Herron-Thorpe et al. (2014) noted that careful consideration must also be given to the
vertical coordinates across models and satellite products, as discrepancies in reporting heights in reference to sea level,
ground level, or the geoid can influence analyses.

**Code availability**

WRF-Chem model code is available for download via the WRF website
(https://www2.mmm.ucar.edu/wrf/users/downloads.html). The FINN utility is available for download via the NCAR
Atmospheric Chemistry Observations & Modeling website (http://bai.acom.ucar.edu/Data/fire).

**Data availability**



US Environmental Protection Agency Air Quality System Data Mart (internet database) is available for download
(https://www.epa.gov/airdata). NCDC data is available for download via the NCEI website
(https://www.ncei.noaa.gov/metadata/geoportal/rest/metadata/item/gov.noaa.ncdc:C00684/html#). TROPOMI data is
available for download via the Copernicus Open Access Hub website (https://scihub.copernicus.eu/). ERA5 data is
available for download via the Copernicus Climate Data Store website (https://cds.climate.copernicus.eu/). FINN
emission data is available for download via the NCAR Atmospheric Chemistry Observations & Modeling website
(http://bai.acom.ucar.edu/Data/fire).

## Author contribution

Y.W., J.H.S., and J.H.J conceived and designed the research. Y.W., and B.R. performed the WRF-Chem simulations.
B.R., Y.W., and J.H.S. performed the data analyses and produced the figures. B. Z. provided technical support for fire
emission preparation. Z.C.Z. helped satellite data analyses. B.R., Y.W., and J.H.S wrote the paper. All authors
contributed to the scientific discussions and preparation of the manuscript.

## Competing Interests

The authors declare that they have no conflict of interest.

## Acknowledgements

This study was supported by the Jet Propulsion Laboratory, California Institute of Technology, under contract with
NASA. We thank Kristal R. Verhulst, Yi Yin, Don Longo, Gonzalo Ferrada, and Saulo Freitas for their support and
discussion.

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

Table 1. Summary of sensitivity simulation setup.

| Name | Fire Data | Smoldering Emissions | Flaming Factor | Entrainment Constant | LSM | Aerosol Radiative Feedback |
|---|---|---|---|---|---|---|
| **S_CTRL₁** | VIIRS | x3 Nov. 13, x2 Nov. 14-16 | Native | 0.05 | Noah/MYJ | Yes |
| S_EMRAW | VIIRS | **Native** | Native | 0.05 | Noah/MYJ | Yes |
| S_NOAERO | VIIRS | x3 Nov. 13, x2 Nov. 14-16 | Native | 0.05 | Noah/MYJ | **No** |
| S_FCTX2 | VIIRS | x3 Nov. 13, x2 Nov. 14-16 | **x2** | 0.05 | Noah/MYJ | Yes |
| S_ENTR | VIIRS | x3 Nov. 13, x2 Nov. 14-16 | Native | **0.02** | Noah/MYJ | Yes |
| S_LSM | VIIRS | x3 Nov. 13, x2 Nov. 14-16 | Native | 0.05 | **P-X/ACM2** | Yes |



| S_FINN | **FINN** | - | - | 0.05 | Noah/MYJ | Yes |

[1]Scenario that agrees best with surface observations and is of primary focus in this study. Bold denotes parameter perturbed from the S_CTRL scenario.





Table 2. Summary of meteorological model performance metrics for the simulation duration.

| Variable | Parameter | Near Source[1] | Sacramento[1] | Bay Area[1] | Station 27 | Station 28 |
|---|---|---|---|---|---|---|
| Wind Speed[2] ($m\ s^{-1}$) | Observation Mean | 1.4 (0.2) | 1.0 (0.2) | 1.6 (0.7) | 1.5 | 0.7 |
| | S_CTRL Mean | 2.6 (0.3) | 1.4 (0.4) | 2.0 (0.7) | 2.3 | 1.0 |
| | Mean Bias | 1.2 | 0.5 | 0.5 | 0.9 | 0.3 |
| Wind Direction[3] (deg) | Observation Mean | 360.0 | 338.2 | 73.9 | 68.9 | 148.8 |
| | S_CTRL Mean | 356.9 | 325.9 | 26.7 | 72.8 | 11.8 |
| | Mean Bias | 2.9 | 11.0 | 0.2 | 2.8 | 10.1 |
| Temp (°C) | Observation Mean | 8.2 (2.3) | 10.1 (1.7) | 10.8 (1.9) | 9.9 | 8.1 |
| | S_CTRL Mean | 12.5 (3.6) | 13.7 (1.4) | 15.7 (1.2) | 15.5 | 13.8 |
| | Mean Bias | 4.4 | 3.6 | 4.9 | 5.6 | 5.7 |

[1]Area winds are averaged for 4 stations near source, 6 stations in Sacramento, and 12 stations in the Bay Area. Area temperatures are averaged for 10 stations near source, 7 in Sacramento, and 13 in the Bay Area. Standard deviation of station averages is noted in parenthesis.
[2]Mean wind speed is calculated as the average of the magnitude of the wind vector.
[3]Mean wind direction is calculated assuming a unity vector.


Table 3. Summary of model performance metrics for surface PM$_{2.5}$ ($\mu$g m$_{-3}$) for the simulation duration.

| Parameter | Near Source* | Sacramento* | Bay Area* | Station 27 | Station 28 |
|---|---|---|---|---|---|
| Observation Mean | 98.3 (39.7) | 77.2 (24.9) | 74.1 (5.4) | 77.9 | 69.8 |
| S_CTRL Mean | 163.1 (108.5) | 65.8 (16.3) | 57.2 (6.4) | 68.1 | 63.6 |
| Mean Bias | 64.8 | -11.4 | -16.8 | -9.9 | -6.2 |
| Normalized Mean Bias | 76.5% | -17.4% | -23.1 | -12.7% | -8.9% |

*Area values are averaged for 5 stations near source, 7 stations in Sacramento, and 13 stations in the Bay Area. Standard deviation of station averages is noted in parenthesis.




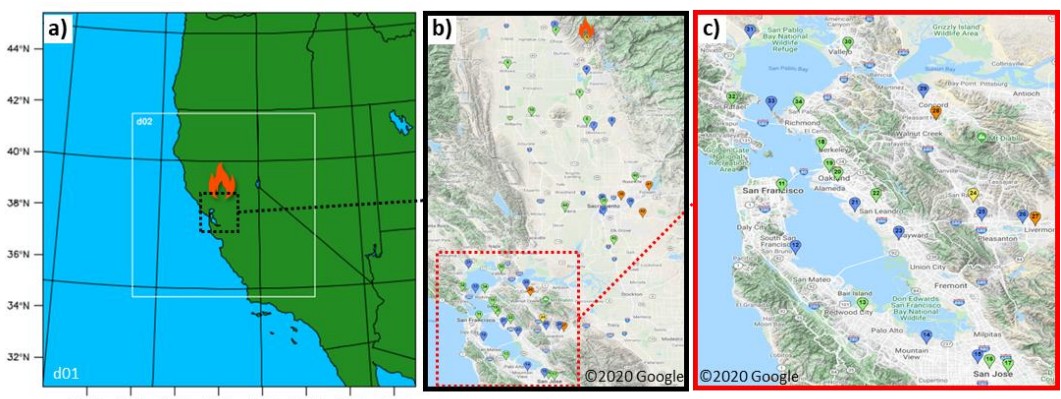

Figure 1. Study domain (a) and observation station locations (b,c). Domain d01 covers the western US with a horizontal resolution of 8 km. Domain d02 is centered over northern California with a horizontal resolution of 2 km. AQS and NCDC observation sites are shown in panel b and panel c, where stations marked in green measure only PM$_{2.5}$, stations in blue measure wind and temperature, stations in orange measure both PM$_{2.5}$ and meteorology, and stations in yellow measure temperature only. Additionally, BC and CO are measured at 8 and 12 sites in the Bay Area, respectively.



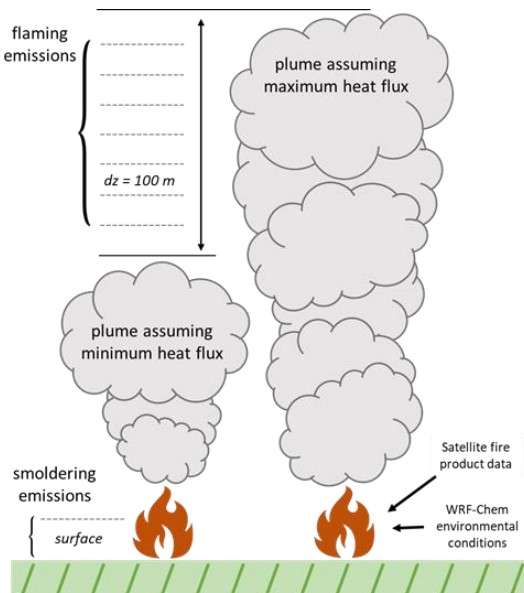

Figure 2. Plume rise model schematic. For each grid cell in which wildfire occurs, the plume rise model uses satellite fire products and the surrounding WRF-Chem environmental conditions to calculate two plume top heights by using the land-type dependent minimum and maximum wildfire heat fluxes. Smoldering phase emissions are allotted to the surface layer, while flaming phase emissions are distributed linearly aloft within the injection layers at a vertical resolution of 100 m.




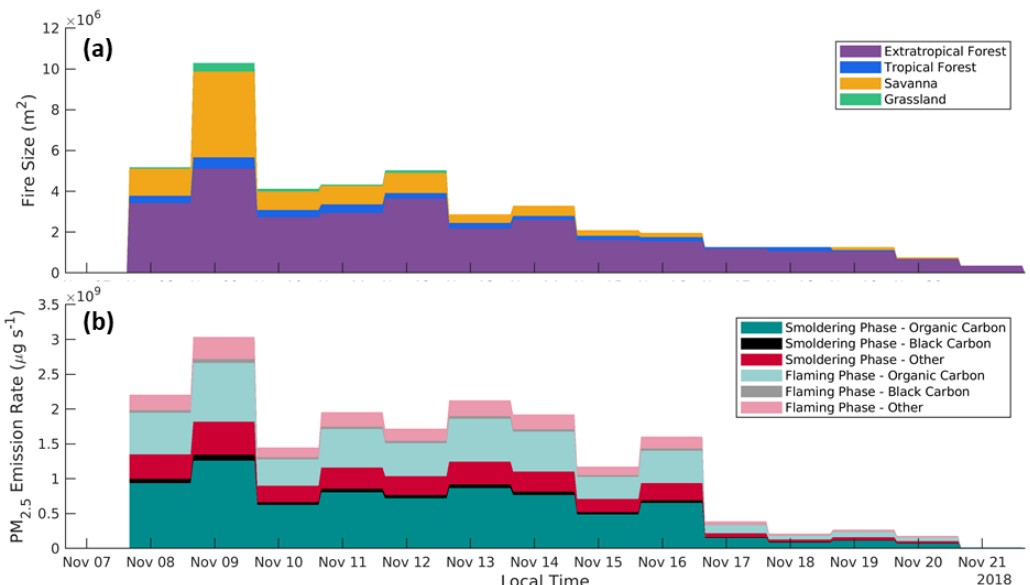

Figure 3. Wildfire area by vegetation type in m2 (a) and PM2.5 emission rate in µg s-1 by combustion phase and species
(b) input into WRF-Chem. The base inventory is produced from VIIRS and MODIS fire products using the PREP-
CHEM-SRC processor and is employed by S_EMRAW. The control and remaining sensitivity simulations use an
inventory with triple emission flux of all species on 13 November and double during 13-16 November, shown here.
About 59% of total PM2.5 emissions occur in the smoldering phase (darker colors in panel b). The total PM2.5 emitted
is composed of 69.5% organic carbon and 4.5% black carbon. The Camp Fire burned primarily extratropical forest
(purple) followed by savanna (yellow). Burning of extratropical forest generated the greatest fraction of emissions in
the flaming phase at 44.2%, followed by savanna at 22.9% and tropical forest at 17.4%. Grassland emits only in the
smoldering phase.

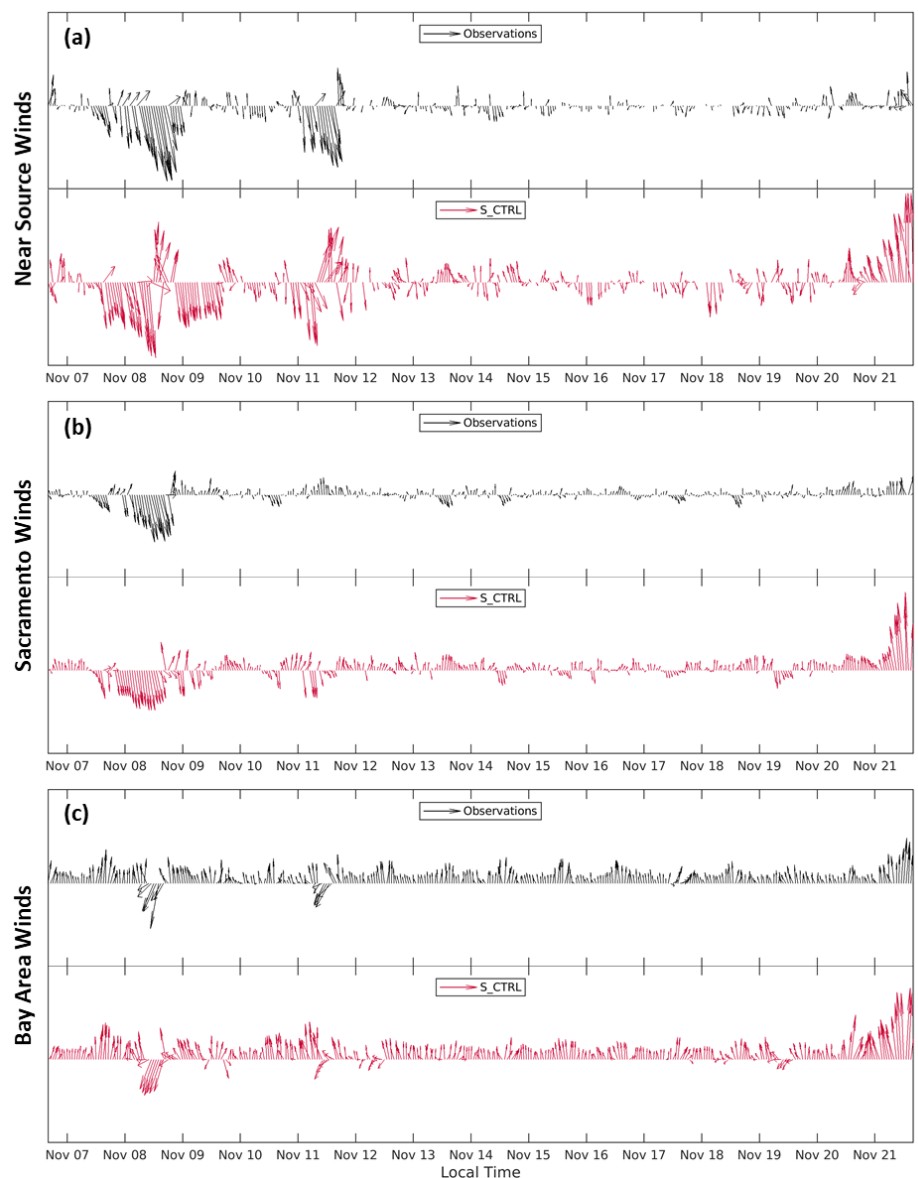

Figure 4. Comparison of AQS and NCDC wind observations (black) with S_CTRL predictions (red) averaged over the three areas of study: a) near the wildfire (N = 4), b) Sacramento (N = 6), and c) the Bay Area (N = 12). Arrows indicate the wind direction and their length represents wind speed. For reference, S_CTRL predicts maximum wind speeds of 8.7, 7.5, and 7.1 m s-1 near the source, in Sacramento, and in the Bay Area, respectively. Paradise and the Sacramento areas experienced strong northerly winds during the first few days of the fire. S_CTRL generally predicted
faster and more variable winds, but broader trends in Sacramento and the Bay Area were represented well.



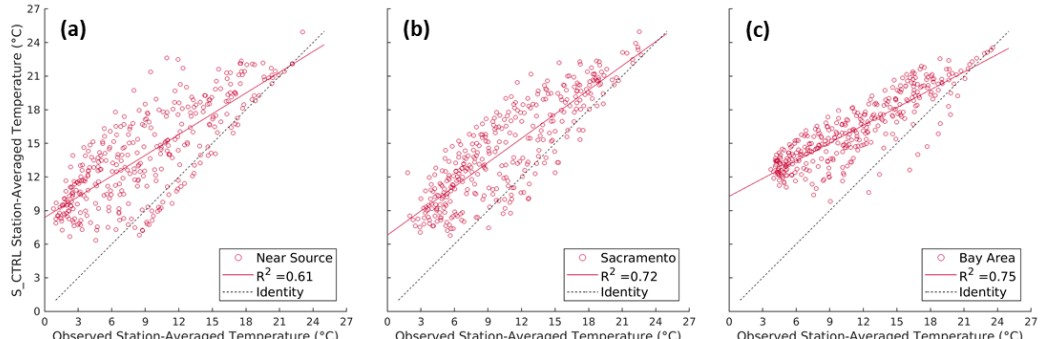

Figure 5. Comparison of AQS and NCDC temperature observations versus S_CTRL predictions: a) near the wildfire (N = 10), b) Sacramento (N = 7), and c) the Bay Area (N = 13). The solid red lines show a linear regression fit, while the dotted black lines denote 1:1 simulations vs. observations. The simulations achieved a correlation coefficient $R_2$ of 0.61 near the fire, 0.72 in Sacramento, and 0.75 in the Bay Area.




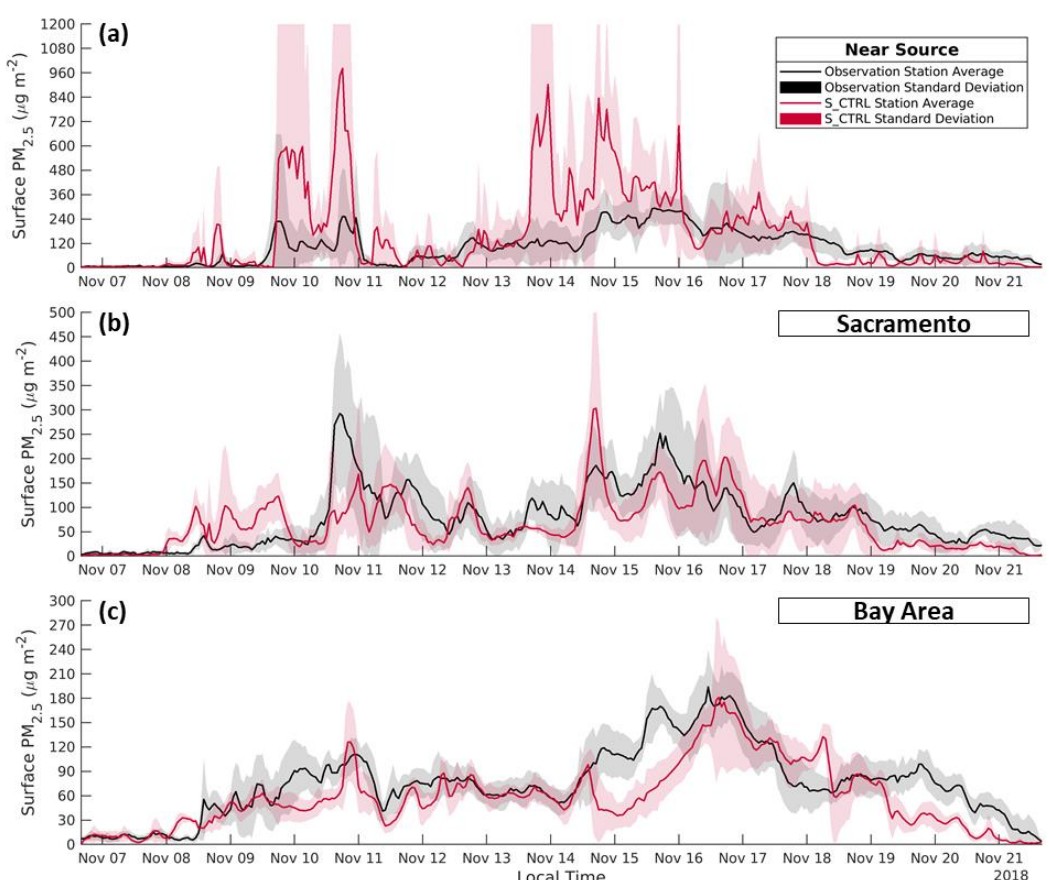

Figure 6. Comparison of AQS surface PM2.5 observations (black) with S_CTRL predictions (red) averaged over the three areas of study: a) near the wildfire (N = 5), b) Sacramento (N = 7), and c) the Bay Area (N = 13). Shading indicates the standard deviation of the sampled stations. S_CTRL overpredicted PM2.5 in the region in the vicinity of the fire but performed well in the areas downwind.


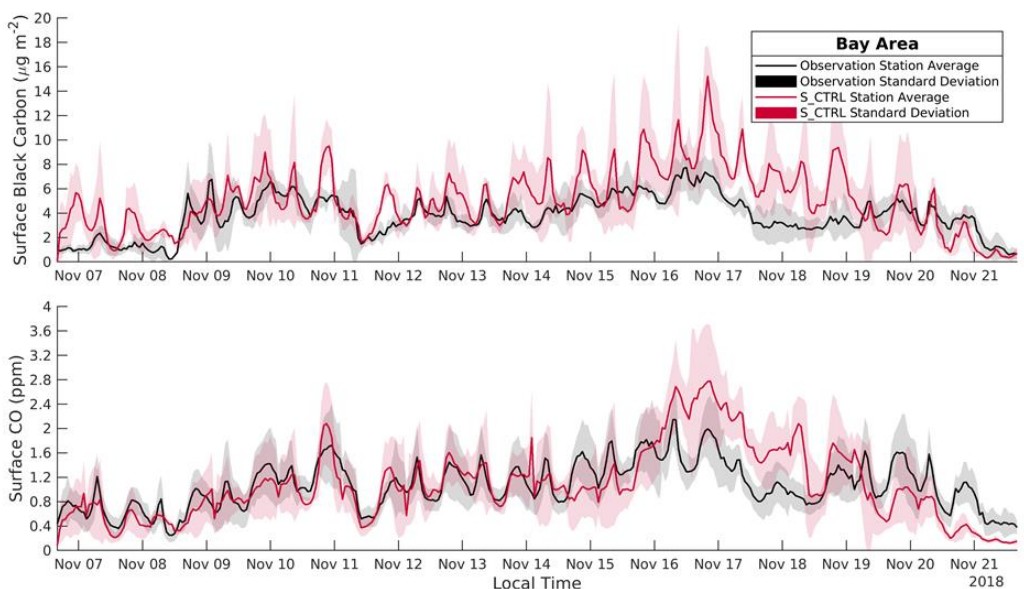

Figure 7. Comparison of AQS surface black carbon (a, N = 5) and carbon monoxide (b, N = 12) observations (black)
with S_CTRL predictions (red) at monitoring sites in the Bay Area. S_CTRL captures the temporal evolution of BC
and CO and is close to observed values. BC peaks are often overpredicted. The greatest bias of BC and CO occurs
during 16-18 November, likely due to the scale factor applied to emissions during 13-16 November.

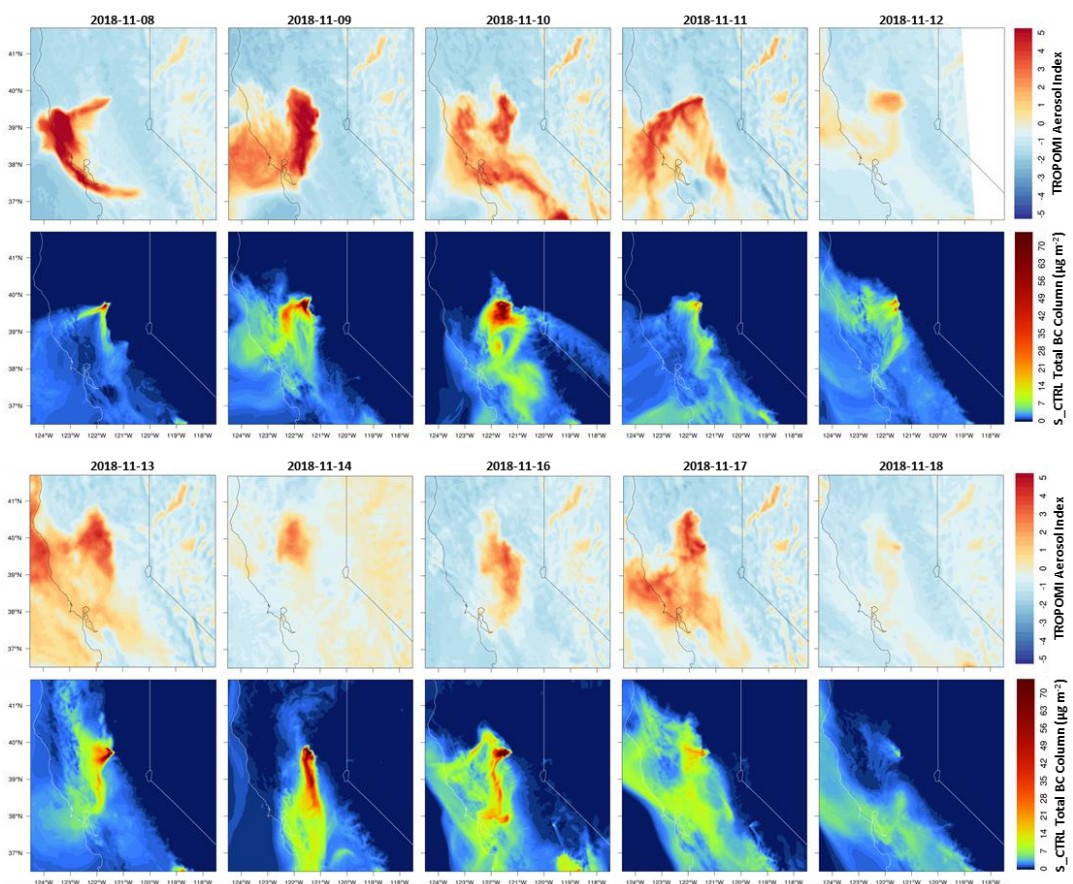

Figure 8. Comparison of TROPOMI UV aerosol index and S_CTRL total BC column during 8-18 November at 13:30 local time as a proxy for plume structure and motion. Due to cloud coverage, no data for 15 November are shown. Positive aerosol index (warm colors) indicates aerosols that absorb radiation like black and brown carbon. The spatial distribution of the plume is generally captured on most days. The simulation also captures some of the finer structures seen by the satellite, though somewhat displaced.



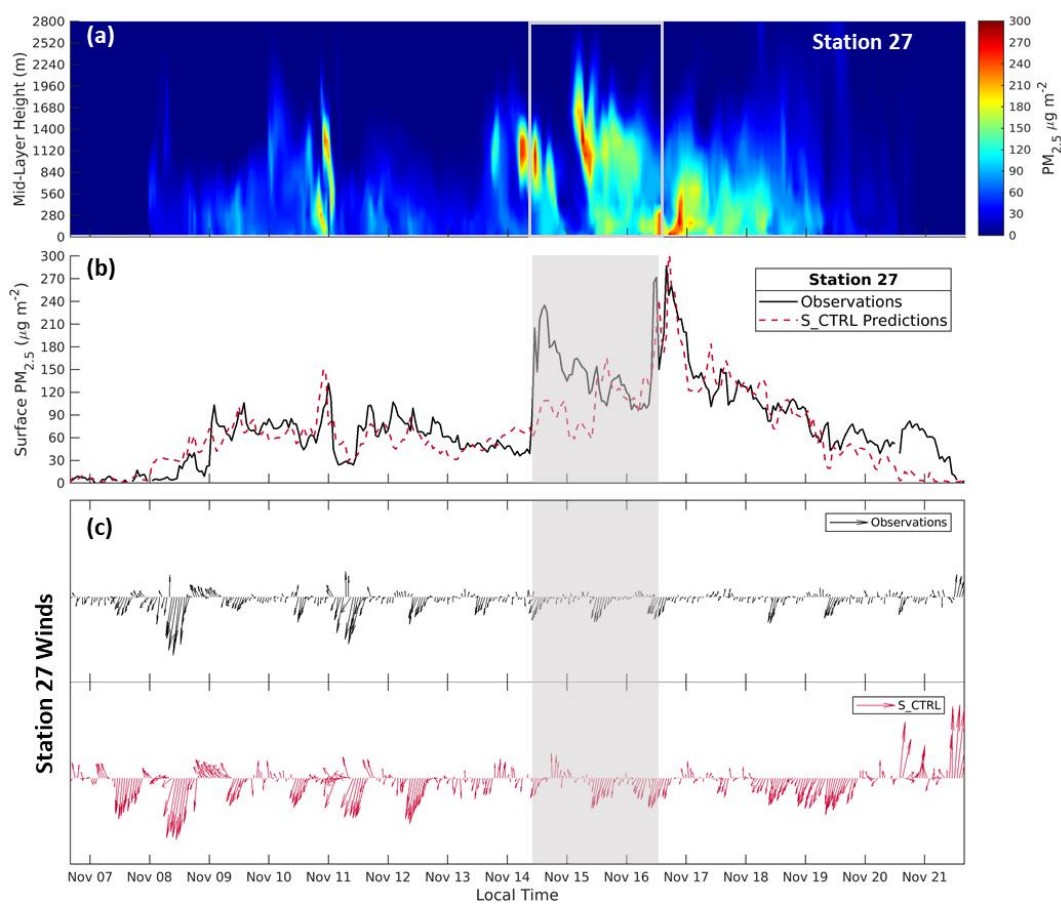

Figure 9. Vertical profile of PM$_{2.5}$ (a), time series of surface PM$_{2.5}$ (b), winds (c; observations in black and predictions in red) at Station 27 in the Bay Area. The gray box highlights the timeframe of greatest model bias of surface PM$_{2.5}$. Sharp increases in PM$_{2.5}$ correlate with a switch to northeasterly winds that import fire emissions to the Bay Area. Large negative PM$_{2.5}$ bias on 15 November occurs when S_CTRL deviates from observations and produces southerly winds which bring in clean air. This can be seen with the column of low level of PM on 15 November in (a).




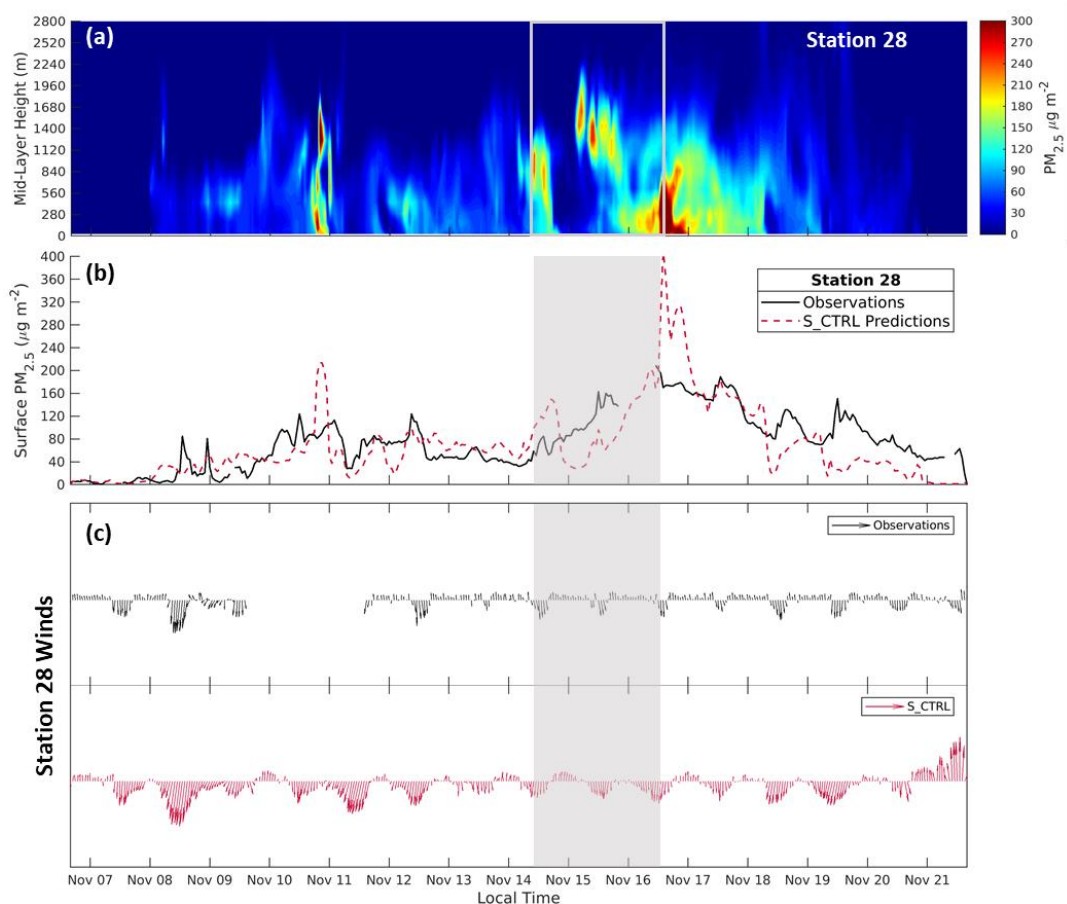

Figure 10. Vertical profile of PM2.5 (a), time series of surface PM2.5 (b), winds (c; observations in black and
predictions in red) at Station 28 in the Bay Area. This station experienced different wind and PM evolution
compared to Station 27 in Figure 10. The gray shading highlights the timeframe of greatest model bias of surface
PM2.5. Sharp increases in PM2.5 correlate with a switch to northerly winds that import emissions to the Bay Area.
Large negative PM2.5 bias on 15 November occurs when S_CTRL deviates from observations and produces stronger
southerly winds. This can be seen with the column of reduced particulate matter on 15 November in (a).






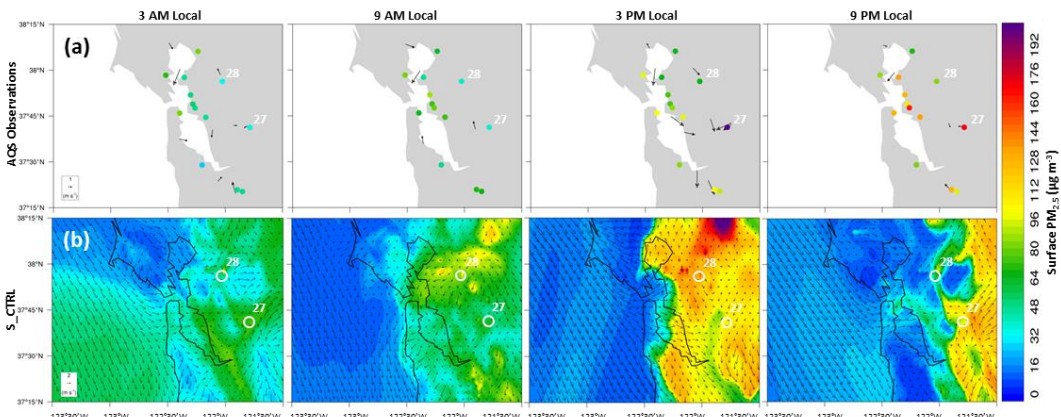


Figure 11. Surface PM$_{2.5}$ and wind field on 14 November in the Bay Area of observations (a) and S_CTRL predictions (b). Note that the reference wind vector for S_CTRL is 2 m s$^{-1}$ while the reference is 1 m s$^{-1}$ for observations. While the plume encroaches on the Bay Area, a strong sea breeze develops midday, driving plumes back inland. This sea breeze is not present in observational data, leading to a large underprediction of surface PM$_{2.5}$.




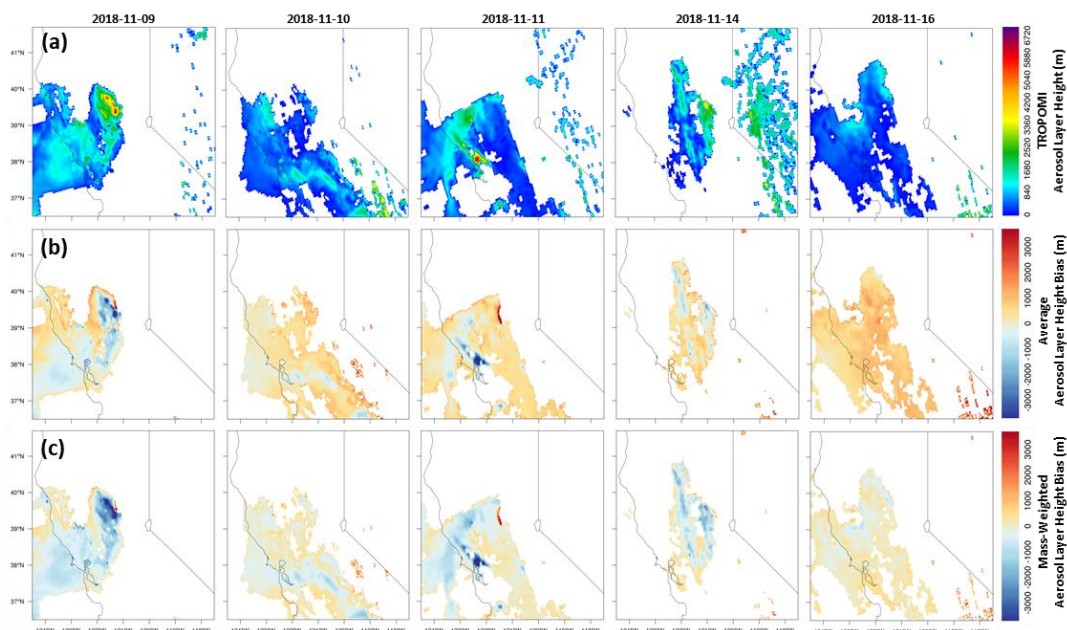

Figure 12. Comparison of TROPOMI aerosol layer height (a) and bias where S_CTRL layer height is calculated as the average of heights where $PM_{2.5} > 3 \, \mu g \, m_{-3}$ (b) and the average weighted by $PM_{2.5}$ mass (c) for select days at 13:30 local time. In panels b and c, warm colors indicate positive bias where S_CTRL overpredicts the height of the aerosol layer.


Figure 13. Comparison of meteorology generated by S_CTRL (solid red) and S_NOAERO (in which aerosol effects do not feed back to the meteorology, dashed blue) over the three areas of study: a) near the wildfire, b) Sacramento, and c) the San Francisco Bay Area. Exclusion of the aerosol feedback has the greatest effect nearest the fire, where S_NOAERO increased wind and temperature by 9.8% and 9.7%, respectively, on average. The aerosol feedback mechanism has the least significance in the Bay Area, where S_NOAERO wind speed differs less than 2% and temperature differs 3.1% on average. The most pronounced changes occur during 14-16 November when S_CTRL significantly underpredicts surface PM$_{2.5}$. In WRF-Chem, the feedback of aerosol-radiation interactions on meteorology act to stabilize the atmosphere, slow wind speeds, and increase PM concentrations.



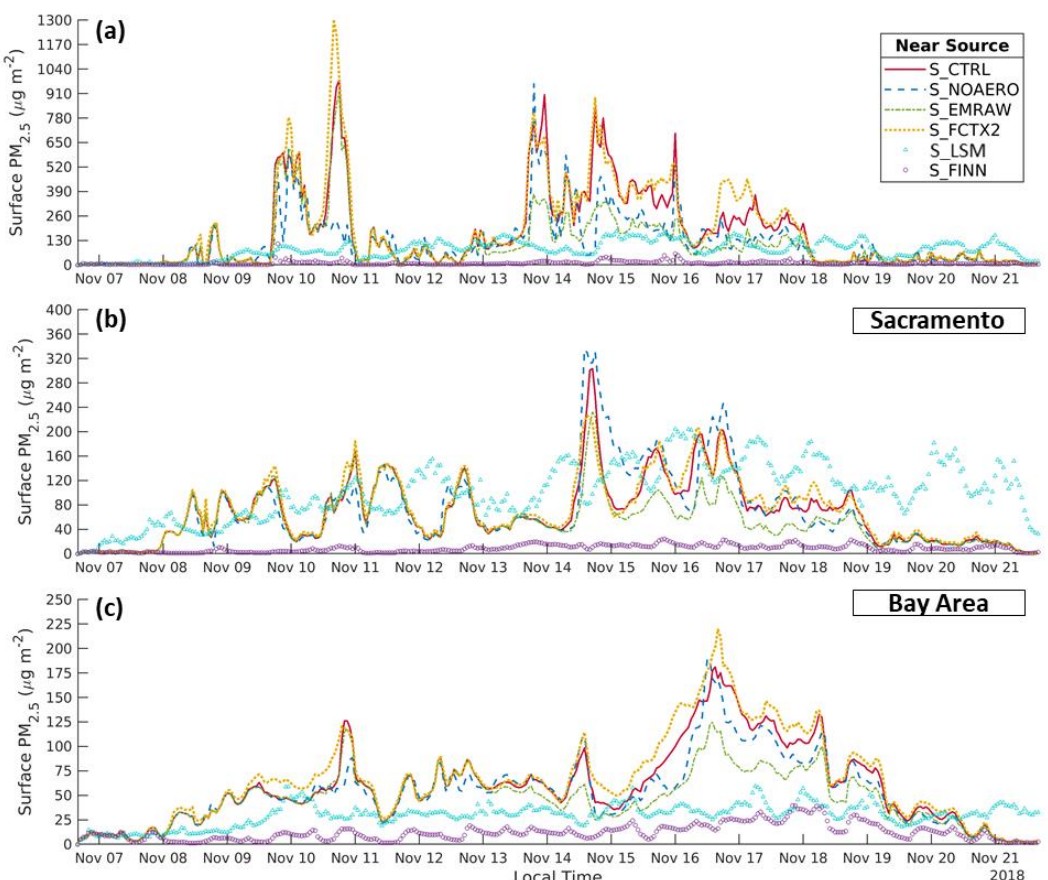


Figure 14. Time series of surface PM$_{2.5}$ (µg m$_{-3}$) predicted by the sensitivity simulations (Table 1) averaged for the three areas of study: a) near the wildfire (N = 5), b) Sacramento (N = 7), and c) the Bay Area (N = 13). S_ENTR is omitted from the figure as it resulted in less than 1% change from S_CTRL. In the Bay Area, S_FCTX2 generally predicted more surface PM$_{2.5}$, recovering 10-35 µg m$_{-3}$ 14-16 November when S_CTRL significantly underpredicts

PM$_{2.5}$ compared to observations. S_EMRAW demonstrates the impact of increasing the emissions inventory for 13-16 November. In the Bay Area, using the unperturbed emissions inventory reduces PM$_{2.5}$ by more than 30% over 14-16 November. The impact of the aerosol feedback mechanism on PM$_{2.5}$ (S_NOAERO) is location dependent. Excluding the feedback to meteorology generally reduces PM$_{2.5}$ near the wildfire and in the Bay Area, while increasing PM$_{2.5}$ in Sacramento. Employing the ACM2 PBL scheme results in a vastly different temporal evolution with a distinct

diurnal pattern (S_LSM). FINN input fire data produces very little PM$_{2.5}$.