# Peer review of "Air Quality Impact of the Northern California Camp Fire of November 2018"

_Atmospheric Chemistry and Physics, 2020_

## Referee Comment (RC1) · Anonymous Referee #1 · 31 Jul 2020

This paper presents the results of a systematic aerosol dispersion and air quality modeling exercise based on the 2018 Camp Fire in northern California. A careful analysis is given, using a combination of satellite and surface measurements over the duration of the fire. There is a great need to analyze and improve such models, to provide reliable air quality forecasting. This work is worthy of publication in ACP in my opinion. I hope the notes below offer some avenues for minor improvements.

1. Lines 110 to 116. I'm not clear what values were selected for some of the parameters in Equation 1, and the degree to which uncertainty in these parameters affects the final model results. This is only partly addressed in Section 4.2; presenting what was learned about emissions process modeling in more detail might be helpful.

2. Lines 122-123. Wildfires tend to have a very distinct diurnal cycle. Especially

given the extensive modeling effort performed here, using a 13:30 local time sample as diurnally representative might not be the best assumption. There also might be MODIS FRP data at 10:30 am as well as nighttime sampling for this fire.

3. Lines 144-146. The Camp Fire reportedly also burned the town of Paradise, California between 8 and 10 November 2018. Does urban structure represent a land cover type that should be included in the simulation, as it can produce very different emissions from grassland or forest?

4. Lines 274-278. Given the assumptions required to perform the TROPOMI ALH retrieval, it might be worth comparing the results with any height retrievals from MODIS/MAIAC (Lyapustin et al. 2019, doi:10.1109/LGRS.2019.2936332) or MISR or CALIPSO. The comparison might help quantify measurement uncertainty. Line 285. Is the TROPOMI ALH actually accurate to 100 m?

5. Line 282. Figure 9 is first referenced after Figures 10, 11, and 12. Probably warrants re-numbering.

6. Lines 337-339. There are notable uncertainties in the satellite estimates of smoke emissions. For example, the satellite results are not species-specific, relatively coarse pixel resolution contributes, etc. The factor of 5 adjustment from Archer-Nicholls et al. is not unusual, and is an indication of the underlying limitations.

7. Given the complexity of the problem, I understand why you perturb individual factors in this study. As you have built an advanced modeling capability to assess smoke dispersion for air quality applications, I'm wondering whether testing at least a few combinations of the main factors might yield some additional insights. There are likely some non-linear interactions among mechanisms, and if the goal is to improve air quality prediction, this might be important.

---

## Referee Comment (RC2) · Anonymous Referee #2 · 6 Aug 2020

A comprehensive study on surface-level air quality impacts of the 2018 Camp Fire is conducted using a combination of WRF-Chem numerical simulations, ground-based monitoring station observations of PM2.5, black carbon ,carbon monoxide and meteo-rology, and a suite of space-borne satellite measurements for three separate regions, including (i) close proximity to the fire, (ii) the Sacramento Metropolitan Area and (iii) the San Francisco Bay Area. Evaluation of model simulations against ground-based observations showed good agreement for surface-level wind fields, ambient tempera-tures, and temporal trends in downwind PM2.5 and black carbon concentrations. Com-parison to satellite products demonstrated the ability of model simulations to replicate the general spatiotemporal structure and evolution of the wildfire plumes. Sensitivity analyses were performed to investigate the influence of key parameter perturbations

on the accuracy of model predictions relative to the baseline control simulation, including (i) aerosol radiative feedback, (ii) boundary layer dynamics, (iii) plume rise and entrainment, (iv) fire inventory data, (v) emission rates and (vi) flaming versus smoldering partitioning. Results indicate greatest sensitivity to fire emission and boundary layer parameterizations. The main objective of these efforts is to assist improvement in air quality forecasting of wildfire events to ultimately protect human health and reduce economic impact.

Major Comments: The authors do a commendable job in the scale and scope of their simulations and analyses. There is little doubt these efforts will be of interest to the broader community and promote forward movement of this field. The reviewer recommends publication upon consideration of a few key points and minor revisions.

1. Perhaps the most striking feature of these wildland-urban interface firestorms is the scale of destruction of the built environment, including Santa Rosa during the 2017 Napa/Sonoma wildfires, Redding during the 2018 Carr Fire, and Paradise during the 2018 Camp Fire studied here. Although wildfires have been studied for decades and there is vast literature characterizing biomass combustion emissions, there are large knowledge gaps in the composition and toxicity of these emissions when a nontrivial fraction of the burnt area includes built environment comprising a vast array of non-biomass related materials. There is clearly a paucity of the types of land cover and fire emissions data required to incorporate these considerations into model simulations, but the reviewer feels it is a key point of sufficient significance to merit inclusion in the manuscript, if only from a speculative perspective. This discussion could easily be incorporated into section 4.2 – Fire Emission Inventory – or as a standalone subsection. Is it possible to calculate what fraction of the burned area can be attributed to the built environment relative to the other landcover vegetation types for the days that Paradise burned? If so, then these data could be included in Figure 3. Presumably, a large fraction of the non-biomass related materials do not sustain flaming combustion but rather are subjected to high temperature pyrolysis analogous to smoldering, which

impacts gas-particle partitioning, particle size and composition, and injection heights, and thus downwind simulated surface-level PM2.5 concentrations. A brief synopsis of these complexities and their impact on model performance would be beneficial.

2. There is a surplus of figures in the manuscript (14 total), many of which are large multi-panel figures, and some effort should be made to condense these to a critical mass necessary for effective visual dissemination of results and conclusions. For example, Figures 1 and 2, although well crafted, are nonessential to reader comprehension and can easily be described in text. Similarly, Figures 4 and 5 do not elucidate additional clarification to what is already discussed in the manuscript and well summarized in Table 2. Furthermore, Figures 10-12 all support the same underlying fundamental conclusion: deviation of simulated wind fields from observation explains underprediction of downwind surface-level PM2.5 mass concentrations in the Bay Area for the period Nov. 14-16. Only one figure (10 or 11) is necessary to make the point.

Minor Revisions: 1. Consider not abbreviating LSM (land surface model) in Table 1; 2. Figures 6, 7, 10, 11, and 14: change y-axis and color scales from ug/m^2 to ug/m^3; 3. In Table 3, Bay Area normalized mean bias is missing percentage symbol (%); 4. Figures should be numbered in the order in which they are discussed within the text, but Figures 10-12 are discussed prior to Figure 9; please renumber figures.
* * *

---

## Author Comment (AC1) · 14 Sep 2020

**Referee 1:**

This paper presents the results of a systematic aerosol dispersion and air quality modeling exercise based on the 2018 Camp Fire in northern California. A careful analysis is given, using a combination of satellite and surface measurements over the duration of the fire. There is a great need to analyze and improve such models, to provide reliable air quality forecasting. This work is worthy of publication in ACP in my opinion. I hope the notes below offer some avenues for minor improvements.

We appreciate the reviewer's valuable comments and constructive suggestions. We have carefully revised the manuscript according to these comments. Point-by-point responses are provided below. The reviewer's comments are in black, our responses are in blue, and the quotes from our manuscript are in italics.

1. Lines 110 to 116. I'm not clear what values were selected for some of the parameters in Equation 1, and the degree to which uncertainty in these parameters affects the final model results. This is only partly addressed in Section 4.2; presenting what was learned about emissions process modeling in more detail might be helpful.

The physical meaning and values of the parameters in the wildfire emission have been further elaborated on Page 5:
"*for a certain species η, $\alpha_{veg}$ is the carbon density (the mass of burnable above-ground biomass per unit area of vegetation), $\beta_{veg}$ is the combustion factor, $EF_{veg}$ is the emission factor by species and vegetation type, and $a_{fire}$ is the burning area of each fire pixel. Vegetation type is generated from the MODIS data following IGBP land cover classification. Vegetation type-specific emission factors ($EF_{veg}$) and combustion factors ($\beta_{veg}$) are derived from Ward et al. (1992) and Andreae and Merlet (2001). Vegetation type-specific carbon density ($\alpha_{veg}$) is based on Olson et al. (2000) and Houghton et al. (2001).*"*

2. Lines 122-123. Wildfires tend to have a very distinct diurnal cycle. Especially given the extensive modeling effort performed here, using a 13:30 local time sample as diurnally representative might not be the best assumption. There also might be MODIS FRP data at 10:30 am as well as nighttime sampling for this fire.

We agree with the reviewer that the diurnal cycle of wildfires can be important in simulating fire-related aerosol formation. However, the current fire emission module used in this study, the "PREP-CHEM-SRC-1.5", averages all fire detections from satellite within one day and does not provide diurnal information, even though the VIIRS satellite provides one daytime retrieval and one nighttime retrieval of active fire count. We have added a statement in the discussion on Page 13 "*Future studies are needed to further improve the present modeling framework to simulate wildfires. Some wildfires exhibit a distinct diurnal cycle, but the current fire preparation module does not utilize the nighttime fire radiative power measurements by the polar-orbiting satellites*".

3. Lines 144-146. The Camp Fire reportedly also burned the town of Paradise, California between 8 and 10 November 2018. Does urban structure represent a land cover type that should be included in the simulation, as it can produce very different emissions from grassland or forest?

We have added a statement in the discussion on Page 13 "*Also, the current land cover and vegetation type data are still relatively coarse in spatial resolution and classification accuracy, which cannot fully resolve a small town in a rural area. In fact, the Camp Fire reportedly burned the town of Paradise, California between 8 and 10 November 2018. This discrepancy definitely contributes to the uncertainty in the fire emission preparation. Additional verification of input fire data sources, such*

*as FINN, and their implementation in the WRF-Chem plume rise model is needed for studies of the vertical structure*".

4. Lines 274-278. Given the assumptions required to perform the TROPOMI ALH retrieval, it might be worth comparing the results with any height retrievals from MODIS/MAIAC (Lyapustin et al. 2019, doi:10.1109/LGRS.2019.2936332) or MISR or CALIPSO. The comparison might help quantify measurement uncertainty. Line 285. Is the TROPOMI ALH actually accurate to 100 m?

We have now clarified that the median error of TROPOMI over land is about 1.75 km (Nanda et al., 2020), so most model-obs. differences are within that retrieval bias. We do not find the plume height products about the Camp Fire from either MODIS/MAIAC or MISR in the public domain, so we leave this satellite intercomparison task for the future study. We have added a statement in the discussion on Page 13 "*The recent TROPOMI aerosol layer height product shows promise as an analytical tool, but requires further development of the method by which it can be directly compared to WRF-Chem. Given the assumptions required to perform the TROPOMI ALH retrieval, more research is needed to compare that product with any height retrievals from MODIS/MAIAC (Lyapustin et al. 2019), MISR (Kahn, 2020), and CALIPSO*".

5. Line 282. Figure 9 is first referenced after Figures 10, 11, and 12. Probably warrants re-numbering.

We have re-ordered the figures as suggested.

6. Lines 337-339. There are notable uncertainties in the satellite estimates of smoke emissions. For example, the satellite results are not species-specific, relatively coarse pixel resolution contributes, etc. The factor of 5 adjustment from Archer-Nicholls et al. is not unusual, and is an indication of the underlying limitations.

We agree with the reviewer on the satellite product uncertainty.

7. Given the complexity of the problem, I understand why you perturb individual factors in this study. As you have built an advanced modeling capability to assess smoke dispersion for air quality applications, I'm wondering whether testing at least a few combinations of the main factors might yield some additional insights. There are likely some non-linear interactions among mechanisms, and if the goal is to improve air quality prediction, this might be important.

To address the reviewer's comment about the non-linearity of different factors in regulating the fire-related PM pollution, we have added a new experiment by jointly perturbing two chosen factors, i.e. emission flaming factor and aerosol radiative feedback. We compare the results from this joint perturbation experiment with those from each individual perturbation experiment and the linear sum of the two. A new Figure 14 is provided with a new paragraph of discussion.

[Figure]

Fig. 14. Comparison of the effects on PM2.5 simulations in the Bay Area from the individual factor perturbation experiments and joint perturbation experiment.

We have added a new discussion on Page 12 "*To test the linearity of different factors in regulating the fire-related PM pollution, we choose two factors, emission flaming factor and aerosol radiative feedback, and conduct a new experiment by jointly perturbing these two. We compare the results from this joint perturbing experiment with those from each individual perturbing experiment and the linear sum of the two in Figure 14. It shows that for the most times, the effect of joint perturbation is close to the sum of the two individual effects (the black line follows well with the black circles), indicating that the relatively good linearity and additivity holds between those two factors in a general sense. The exception occurs under the extreme conditions. During Nov. 14-18 when the plume was thick and PM2.5 concentration was highest in the Bay Area, the aerosol radiative feedback dominates, and the effect of joint perturbation is close to the aerosol radiative effect (the black line follows well with the blue dotted line)*".

We have also added a statement in the conclusion and discussion section on Page 13 "*Given the complexity of the problem, here we mainly perturb individual factors in this study. Future studies can test different combinations of the main factors identified by the present study, which can yield additional insights about non-linear interactions among different processes related with fire emission and transport*".

**Referee 2:**

A comprehensive study on surface-level air quality impacts of the 2018 Camp Fire is conducted using a combination of WRF-Chem numerical simulations, ground-based monitoring station observations of PM2.5, black carbon ,carbon monoxide and meteorology, and a suite of space-borne satellite measurements for three separate regions, including (i) close proximity to the fire, (ii) the Sacramento Metropolitan Area and (iii) the San Francisco Bay Area. Evaluation of model simulations against ground-based observations showed good agreement for surface-level wind fields, ambient temperatures, and temporal trends in downwind PM2.5 and black carbon concentrations. Comparison to satellite products demonstrated the ability of model simulations to replicate the general spatiotemporal structure and evolution of the wildfire plumes. Sensitivity analyses were performed to investigate the influence of key parameter perturbations on the accuracy of model predictions relative to the baseline control simulation, including (i) aerosol radiative feedback, (ii) boundary layer dynamics, (iii) plume rise and entrainment, (iv) fire inventory data, (v) emission rates and (vi) flaming versus smoldering partitioning. Results indicate greatest sensitivity to fire emission and boundary layer parameterizations. The main objective of these efforts is to assist improvement in air quality forecasting of wildfire events to ultimately protect human health and reduce economic impact.

Major Comments: The authors do a commendable job in the scale and scope of their simulations and analyses. There is little doubt these efforts will be of interest to the broader community and promote forward movement of this field. The reviewer recommends publication upon consideration of a few key points and minor revisions.

We appreciate the reviewer's valuable comments and constructive suggestions. We have carefully revised the manuscript according to these comments. Point-by-point responses are provided below. The reviewer's comments are in black and our responses are in blue.

1. Perhaps the most striking feature of these wildland-urban interface firestorms is the scale of destruction of the built environment, including Santa Rosa during the 2017 Napa/Sonoma wildfires, Redding during the 2018 Carr Fire, and Paradise during the 2018 Camp Fire studied here. Although wildfires have been studied for decades and there is vast literature characterizing biomass combustion emissions, there are large knowledge gaps in the composition and toxicity of these emissions when a nontrivial fraction of the burnt area includes built environment comprising a vast array of non-biomass related materials. There is clearly a paucity of the types of land cover and fire emissions data required to incorporate these considerations into model simulations, but the reviewer feels it is a key point of sufficient significance to merit inclusion in the manuscript, if only from a speculative perspective. This discussion could easily be incorporated into section 4.2 – Fire Emission Inventory – or as a standalone subsection. Is it possible to calculate what fraction of the burned area can be attributed to the built environment relative to the other landcover vegetation types for the days that Paradise burned? If so, then these data could be included in Figure 3. Presumably, a large fraction of the non-biomass related materials does not sustain flaming combustion but rather are subjected to high temperature pyrolysis analogous to smoldering, which impacts gas-particle partitioning, particle size and composition, and injection heights, and thus downwind simulated surface-level PM2.5 concentrations. A brief synopsis of these complexities and their impact on model performance would be beneficial.

As suggested by the reviewer, we now discuss the fact that there is a paucity of the types of land cover (especially residential area) and fire emissions data required to incorporate these considerations into model simulations in the beginning of Section 4.2, which serves as motivation to conduct emission perturbation experiments. We have estimated that the area of Paradise, covering 11,614 acres, corresponds to about 7.6% of the total burned area. We acknowledge that this contributes to the uncertainty in the fire emission preparation in the final discussion.

2.  There is a surplus of figures in the manuscript (14 total), many of which are large multi-panel figures, and some effort should be made to condense these to a critical mass necessary for effective visual dissemination of results and conclusions.  For example, Figures 1 and 2, although well crafted, are nonessential to reader comprehension and can easily be described in text.  Similarly, Figures 4 and 5 do not elucidate additional clarification to what is already discussed in the manuscript and well summarized in Table 2. Furthermore, Figures 10-12 all support the same underlying fundamental conclusion: deviation of simulated wind fields from observation explains underprediction of downwind surface-level PM2.5 mass concentrations in the Bay Area for the period Nov. 14-16. Only one figure (10 or 11) is necessary to make the point.

As suggested, we have removed previous Figures 11-12 and retained Figure 10 only. Meanwhile, we prefer to keep Figs. 1 and 2. Fig. 1 shows the model domains that are important information for the reader. Fig. 2 educates the readers about how the plume rise model works, for which there is no previous literature/document available. Figs. 4 and 5 provide useful information about the prevailing winds and their temporal evolution during the Camp Fire, which are closely related with pollutant transport.

Minor Revisions:
1. Consider not abbreviating LSM (land surface model) in Table 1;
2. Figures 6, 7, 10, 11, and 14:  change y-axis and color scales from ug/m^2 to ug/m^3;
3. In Table 3, Bay Area normalized mean bias is missing percentage symbol (%);
4. Figures should be numbered in the order in which they are discussed within the text, but Figures 10-12 are discussed prior to Figure 9; please renumber figures.

We thank the reviewer for identifying these issues. All the above suggested revisions have been made.